# Effects of Purified β-Glucosidases from *Issatchenkia terricola*, *Pichia kudriavzevii*, *Metschnikowia pulcherrima* on the Flavor Complexity and Typicality of Wines

**DOI:** 10.3390/jof8101057

**Published:** 2022-10-09

**Authors:** Wanying Zhu, Wenxia Zhang, Tao Qin, Jing Liao, Xiuyan Zhang

**Affiliations:** College of Food Science and Technology, Huazhong Agricultural University, Wuhan 430070, China

**Keywords:** *I. terricola*, *P. kudriavzevii*, *M. pulcherrima*, purified β-glucosidase, wine

## Abstract

The aim of this study was to investigate the effects of purified β-glucosidases from *Issatchenkia terricola* SLY-4, *Pichia kudriavzevii* F2-24, and *Metschnikowia pulcherrima* HX-13 (named as SLY-4E, F2-24E, and HX-13E, respectively) on the flavor complexity and typicality of wines. Cabernet Sauvignon wines were fermented by *Saccharomyces*
*cerevisiae* with the addition of SLY-4E, F2-24E, and HX-13E; the fermentation process and characteristics of wines were analyzed. The addition of SLY-4E, F2-24E, and HX-13E into must improved the contents of terpenes, higher alcohols, and esters, and decreased the contents of C_6_ compounds and fatty acids, which enhanced the fruity, floral, and taste aspects, reducing the unpleasant green of wines with no significant difference in their appearance. β-glucosidases from different yeast species produced different aroma compound profiles which presented different flavor and quality. F2-24EW had the best effect on flavor and quality of wine followed by SLY-4EW and HX-13EW. These research results can provide references for the use of β-glucosidases from non-*Saccharomyces* yeasts to improve the flavor complexity, typicality, and quality of wines.

## 1. Introduction

Wine is popular with customers for its high nutritional value and health benefits. In 2020, the consumption of wine was about 2.34 × 10^10^ L in the world (OIV, 2021). However, the flavor complexity and typicality of wine fermented by *Saccharomyces cerevisiae* are poor [1], which would affect its competitiveness on the fruit wine market.

Volatile aroma compounds are very important to the flavor of wine. The release of aroma compounds occurs via a sequential hydrolysis mechanism involving several glycosidases [2]. β-glucosidases are the most important flavor enzymes which can hydrolyze non-volatile glycoside compounds to release volatile compounds [3].

Therefore, co-fermentation using non-*Saccharomyces* yeasts with β-glucosidases and *S. cerevisiae* could increase the contents of volatile varietal aroma compounds, which would improve the flavor complexity and typicality of wines [4,5,6,7]. However, the non-*Saccharomyces* yeasts were easily inhibited by *S. cerevisiae* or the vinification environment [8,9,10]. Under this context, more and more researchers found that adding crude extracts of β-glucosidases from non-*Saccharomyces* yeasts into must could significantly hydrolyze non-volatile glycoside compounds to release the volatile compounds to improve the flavor complexity and typicality of wines [3,11,12,13]. Previous research also found crude extracts of β-glucosidases from *I**ssatchenkia terricola* SLY-4, *Pichia kudriavzevii* F2-24, and *Metschnikowia pulcherrima* HX-13 could significantly improve the flavor complexity and characteristics of wines [14]. Recently, several β-glucosidases from non-*Saccharomyces* yeasts were characterized and purified; the addition of purified β-glucosidases from non-*Saccharomyces* yeasts into must could hydrolyze grape-derived aroma precursors, enhancing the aroma of wine [2,15,16,17]. Purified β-glucosidases from *I. terricola* SLY-4 (named as SLY-4E), *P. kudriavzevii* F2-24 (named as F2-24E), and *M. pulcherrima* HX-13 (named as HX-13E) about characterization were studied in our previous research, however, the effects of SLY-4E, F2-24E, and HX-13E on the flavor complexity and typicality of wines are still unclear.

Therefore, to investigate the effects of SLY-2E, F2-24E, and HX-13E on the flavor complexity and typicality of wines, Cabernet Sauvignon wines were fermented by *S. cerevisiae* with adding SLY-4E, F2-24E, and HX-13E into must, respectively. The fermentation kinetics of *S. cerevisiae*, physicochemical properties, volatile aroma compounds, and sensory indicators of the wines were analyzed. The research results can provide references for the use of β-glucosidases from non-*Saccharomyces* yeasts to improve the flavor complexity, typicality, and quality of wines.

## 2. Materials and Methods

### 2.1. Strains and Media

*I**. terricola* SLY-4, *P**. kudriavzevii* F2-24, and *M**. pulcherrima* HX-13 were isolated from a vineyard (Ningxia, China). *S. cerevisiae* was a commercial strain Actiflore^®^ F33 (Laffort, France).

Yeast extract peptone dextrose medium (YPD, 10 g/L yeast extract, 20 g/L peptone, 20 g/L glucose) was used for starter preparation. YPD agar medium (YPDA) was prepared by adding 20 g/L agar into YPD for *S. cerevisiae* cell counting.

Fermentation medium (10 g/L yeast extract, 20 g/L peptone, 20 g/L glucose, 3 g/L NH_4_NO_3_, 4 g/L KH_2_PO_4_, 0.5 g/L MgSO_4_·7H_2_O, 10 mL/L Tween 80) was used for the fermentation of non-*Saccharomyces* yeasts to produce β-glucosidases.

The β-glucosidases from SLY-4, F2-24, and HX-13 were purified by a high-pressure homogenizer, salting out by ammonium sulfate, DEAE-52 anion exchange chromatography, and SephadexG-75 chromatography. The characteristics of the purified β-glucosidases were as follows. The T_opt_ and pH_opt_ of SLY-4E were 55 °C and 5.5. The T_opt_ and pH_opt_ of F2-24E were 45 °C and 6.5. The T_opt_ and pH_opt_ of HX-13E were 50 °C and 5.5. They were stable at 20–30 °C and pH 5–7. In pH 4, the relative activities of SLY-4E, F2-24E, and HX-13E were 53.97%, 46.74%, and 64.79%, respectively.

### 2.2. Laboratory-Scale Fermentation of Wine

200 mL Cabernet Sauvignon must (residual sugar 212.9 g/L, total acidity 3.2 g/L expressed as tartaric acid) in a 250 mL bottle was macerated at 4 °C for 12 h after adding 50 mg/L total SO_2_. Then, wines were fermented at 20 °C by inoculating 10^6^ CFU/mL *S. cerevisiae* and 1 U/L SLY-4E, F2-24E, and HX-13E, which were named as SLY-4EW, F2-24EW, and HX-13EW, respectively. Wine with no β-glucosidase was used as a control. Each experiment was conducted in triplicate. Fermentation ended when the reducing sugar content in the fermentation broth was less than 4 g/L.

### 2.3. Growth and Sugar Consumption Kinetics of S. cerevisiae

The cell number of *S. cerevisiae* were counted through dilution coating on YPD plate, and the residual sugar was determined through the method recommended by OIV every day during wine fermentation (2009).

### 2.4. Analysis of Physicochemical Characteristics and Volatile Compounds in Wines

Alcohol, total acid, and volatile acid were determined through methods recommended by Shi et al. (2019) [6]. The residual sugar content was expressed as glucose (g/L). The total acid content was expressed as tartaric acid (g/L), and the volatile acid content was expressed as acetic acid (g/L). Each experiment was conducted in triplicate.

The volatile compounds from wines were extracted by headspace solid-phase micro-extraction with 50/30 µm DVB/CAR/PDMS fiber (Supelco, Bellefonte, PA, USA) and analyzed with an Agilent 6890 N gas chromatograph with a DB-5 capillary column (30 m × 0.32 mm × 0.25 µm) coupled to an Agilent 5975B mass spectrometer (GC-MS) [3] with little modification. A total of 8.0 mL wine, 2.0 g NaCl, and 10µL 450 µg/L cyclohexanone (internal standard) were added into a 20 mL headspace bottle and incubated at 40 °C for 15 min with magnetic stirring. The fiber was pushed into the headspace of the bottle for 30 min and immediately desorbed in the injector of GC at 250 °C for 5 min. The analysis condition of GC was as follows: increasing temperature from 40 °C to 130 °C at 3 °C/min, and then to 250 °C at 4 °C/min. The temperatures of injector and detector were set at 250 °C and 260 °C, respectively. The MS was operated in electron impact ionization mode at 70 eV, and ion source temperature was 250 °C. The volatile compounds were identified by comparing the MS fragmentation pattern of each compound with that in database Wiley 7.0 and NIST05. The following formula was used to calculate the content of compounds:$$\mathrm{Compound}\;\mathrm{content}\;\left(µg/\mathrm{mL}\right)=\frac{\mathrm{GC}\;\mathrm{peak}\;\mathrm{areas}\;o\;f\;\mathrm{the}\;\mathrm{compound}\;\times \mathrm{Quality}\;o\;f\;\mathrm{internal}\;\mathrm{standard}\;\left(µg\right)\;}{\mathrm{GC}\;\mathrm{peak}\;\mathrm{area}\;o\;f\;\mathrm{the}\;\mathrm{internal}\;\mathrm{standard}\;\times \mathrm{Volume}\;o\;f\;\mathrm{the}\;\mathrm{sample}\;\left(\mathrm{mL}\right)}$$

### 2.5. Sensory Evaluations of Wine

The sensory evaluation was performed as described by Shi et al. (2019) [6]. Wines were evaluated by ten well-trained panelists (five females and five males) in a tasting room at 20 °C. Approximately 20 mL wine samples were poured into wine glasses and presented in triplicate. Potable water was provided for rinsing the palate during testing. Sensory descriptions, including appearance, aroma (fruity, floral, and green), and taste of wine, were scored from zero (weak) to nine (intense), respectively.

### 2.6. Data Analyses

Microsoft Office 2016 and GraphPad Prism 6.0 were used to complete the data and charts. SPSS 19.0 software (SPSS Inc., Chicago, IL, USA) was used to do one-way analysis of variance (ANOVA) and multiple mean comparisons were completed by the Duncan test. SIMCA-P 14.1 (Umetrics AB, Umea, Sweden) was used for principal component analysis (PCA) of volatile aroma components. MultiExperiment Viewer 4.9.0 (TIGR, Sacramento, CA, USA) was used for hierarchical clustering and heat map visualization of fermentative aroma compounds from wines after the z-score standardization.

## 3. Results and Discussions

### 3.1. Growth and Sugar Consumption Kinetics of S. cerevisiae during Wine Fermentation

The growth and sugar consumption kinetics of *S. cerevisiae* indicated that *S. cerevisiae* could grow normally during wine fermentation. Compared with the control (8.91 × 10^7^ CFU/mL), the maximum biomass of *S. cerevisiae* in SLY-4EW (1.06 × 10^8^ CFU/mL), HX-13EW (1.26 × 10^8^ CFU/mL), and F2-24EW (1.04 × 10^8^ CFU/mL) were higher (Figure 1). The fermentation periods were 7 days with no significant difference among wines. The results indicated that adding SLY-4E, F2-24E, and HX-13E into must was beneficial for the growth of *S. cerevisiae,* which could ensure successful wine fermentation, but it had no effect on the fermentation periods of wines. Zhang et al. (2020) [14] reported that adding crude extracts of β-glucosidase from *I. terricola*, *P. kudriavzevii,* and *M. pulcherrima* into must could increase the maximum biomass of *S. cerevisiae*. However, Belda et al. (2015) [18] and Hu et al. (2020) [19] reported that the maximum biomass of *S. cerevisiae* was decreased during co-fermentation with *Torulaspora delbrueckii*, *Hanseniaspora opuntiae*, and *Hanseniaspora uvarum*, respectively. These results indicated that adding β-glucosidases into must could enhance the maximum biomass of *S. cerevisiae* and had no significant effect on the fermentation periods of wines, but the co-fermentation of yeasts with β-glucosidase activity would decrease the maximum biomass of *S. cerevisiae* and prolong the fermentation periods. The increase in maximum biomass of *S. cerevisiae* might be explained by the following. Adding β-glucosidases could cause the hydrolysis of glycosides to glucose which was used as a carbon source [20], while in co-fermentation, the maximum biomass of the yeasts might be decreased due to competition.

### 3.2. The Physicochemical Characteristics and the Volatile Aroma Compounds of Wines

The content of residual sugar (3.83–4.00 g/L), alcohol (11.62–11.99% *v*/*v*), total acid (6.56–6.75 g/L), and volatile acid (0.27–0.29 g/L) of SLY-4EW, F2-24EW, and HX-13EW had no significant differences (Table 1).

The detected 58 kinds of volatile aroma compounds were categorized into varietal aroma compounds and fermentative aroma compounds. Eleven variety aroma compounds were clustered into C_6_ compounds and terpenes. Forty-seven fermentative aroma compounds were clustered into higher alcohols, fatty acids, fatty acid ethyl esters, acetic esters, and carbonyl compounds (Table 2).

### 3.3. Varietal Aroma Compounds

Eleven varietal aroma compounds were classified into C_6_ compounds and terpenes. The total content of varietal aroma compounds in SLY-4EW (1.98 mg/L), HX-13EW (1.88 mg/L), and F2-24EW (1.64 mg/L) was significantly higher than that in the control (1.41mg/L).

The content of C_6_ compounds with unpleasant green flavor presented a significant decrease in SLY-4EW (0.91 mg/L), F2-24EW (0.84 mg/L), and HX-13EW (0.82 mg/L) compared with that in the control (0.98 mg/L), and the content of terpenes in SLY-4EW (1.07 mg/L), HX-13EW (1.04 mg/L), and F2-24 EW (0.82 mg/L) was significantly higher than that in the control (0.43mg/L). The odor active varietal aroma compounds (OAV > 1) were linalool, citronellol, 1-octen-3-ol, geraniol, and caryophyllene. These results indicated that adding SLY-4E, HX-13E, and F2-24E could decrease the content of C_6_ compounds and increase the content of terpenes. Qin et al. (2021) [35] and Zhang et al. (2020) [14] also reported that fermentations by *I. terricola*, *P. kudriavzevii*, and *M. pulcherrima* with β-glucosidase activity or adding their crude extracts of β-glucosidase could increase the content of terpenes and decrease the content of C_6_ compounds. In addition, adding purified or crude extracts of β-glucosidases from *H. uvarum*, *Rhodotorula mucilaginosa*, or *Candida easanensis* into must could also increase the content of terpenes and C_6_ compounds in wines [12,13,36]. Ma et al. (2017) [37] reported that adding crude extracts of enzymes (mainly including esterases and β-glucosidases) from *P**ichia fermentans* could increase the content of terpenols and C_6_ compounds. These results indicated that β-glucosidases from different yeasts could increase the content of terpenes but have different effects on C_6_ compounds. A high content of terpenes would enhance the fruity and floral aspects of wines [38] and a low content of C_6_ compounds would decrease the unpleasant green flavor of wines [39,40]. β-glucosidases from yeasts could successfully hydrolyze non-volatile odorless precursors to release the volatile odor compounds to increase the content of terpenes, which would improve the flavor and quality of wines. However, the mechanism of β-glucosidases from different yeasts that had different effects on C_6_ compounds is still unclear. Therefore, in the future, more studies should be carried out to investigate the effects of β-glucosidase from different non-*Saccharomyces* yeasts on C_6_ compounds. Moreover, more non-*Saccharomyces* yeasts with β-glucosidase should be selected for lower producing C_6_ compounds.

The PCA was carried out to reveal the correlation and segregation of varietal aroma compounds from different wines. The results indicated that PC-1 (52%) and PC-2 (36.9%) accounted for 88.9% of the total variance (Figure 2). SLY-4EW was clustered with nerolidol, geraniol, citronellol, linalool, and geraniyl acetone at the negative part of PC-1. HX-13EW was grouped with terpinol, 1-octen-3-ol, and geraniyl acetone at the negative end of PC-1. F2-24EW was clustered with lavenol, caryophyllene, and E-3-hexene-1-ol at the forward end of PC-1. The control was clustered with hexanol at the positive end of PC-1. The results showed that adding SLY-4E, HX-13E, and F2-24E could produce different profiles of varietal aroma compounds: SLY-4E increased the release of nerolidol, geraniol, citronellol, linalool, and geraniyl acetone; HX-13E promoted the release of terpinol, 1-octen-3-ol, and geraniylacetone; while F2-24E promoted the release of lavenol, caryophylene, and E-3-hexene-1-ol. Swangkeaw et al. (2009) [41] showed that adding crude extracts of β-glucosidases from *Hanseniaspora sp*. and *Pichia anomala* into Traminette grape juice could increase the content of limonene and linalool oxide. This implied that different β-glucosidases had diverse substrate specificity to produce various kinds of varietal aroma compounds which would present different varietal aroma. In the future, the effect of different non-*Saccharomyces* yeasts with β-glucosidases on the varietal aroma compounds should be analyzed.

### 3.4. Fermentative Aroma Compounds

The forty-seven fermentative aroma compounds from wines included eleven higher alcohols, five fatty acids, twelve ethyl fatty acids, six acetic acid esters, six other esters, and seven carbonyl compounds. Compared with the control (302.48 mg/L), the content of fermentative aroma compounds in SLY-4EW (488.45 mg/L), HX-13EW (469.92 mg/L), and F2-24EW (476.71 mg/L) was higher (Table 2).

The content of higher alcohols in SLY-4EW (418.89 mg/L), HX-13EW (420.71 mg/L), and F2-24EW (401.33 mg/L) was significantly higher than that in the control (256.97 mg/L) (Figure 3), especially for isoamyl alcohol and benzene ethanol. Previous research also reported that adding crude extracts of β-glucosidases from *P. fermentans*, *H. uvarum*, *Trichosporon asahii*, and *Candida parapsilosis* into must or co-fermentation with *S. cerevisiae* and *Lachancea thermotolerans* or *T. delbrueckii* increased the content of higher alcohols [11,37,42,43]. The results indicated that β-glucosidases from non-*Saccharomyces* yeasts could increase the content of higher alcohols. A proper content of higher alcohols (<300 mg/L) would bring fruity and floral flavors to wines; however, it could be counterproductive when it exceeds 400 mg/L. [33,44,45]. Although higher alcohols could improve the flavor complexity, the high concentration of them in SLY-4EW, HX-13EW, and F2-24EW might have a negative impact on the aroma and flavor. A high concentration of higher alcohols might be explained by the transformation of glucose from the hydrolyzation of glycosides by β-glucosidase or amino acids from β-glucosidase degradation through the Ehrlich pathway [46,47].

Esters were formed by fatty acids and alcohols, while excessive fatty acids would present cheesy, fatty, and rancid notes [48]. The content of fatty acids in SLY-4EW (2.81 mg/L), HX-13EW (2.92 mg/L), and F2-24EW (2.82 mg/L) was lower than that in the control (3.58 mg/L) (Figure 3). Ma et al. (2017) [37] also reported that adding crude extracts of β-glucosidase from *P. fermentans* into must could significantly decrease the content of fatty acids in wines, but other research has indicated that crude extracts of β-glucosidase from *R.mucilaginosa*, *H.uvarum*, *I. terricola*, *P. kudriavzevii,* or *M. pulcherrim* could increase the content of fatty acids in fruit wines [11,14,36]. These results indicated that adding β-glucosidases from different yeasts into must had different effects on the content of fatty acids. However, the reason why β-glucosidases from different yeasts have different effects on the content of fatty acids is unclear.

The concentration of esters was significantly higher in SLY-4EW (58.82 mg/L), HX-13EW (44.96 mg/L), and F2-24EW (70.95 mg/L) than in the control (40.80 mg/L) (Figure 3). Ma et al. (2017) and Hu et al. (2016a) also reported that adding crude extracts of β-glucosidase from *P. fermentans* or *H. uvarum* into must significantly increased the content of esters. These results indicated that adding β-glucosidases from yeasts into must could increase the content of esters, which would present the fruity and floral flavors in wines [13]. The higher content of esters might be explained by the high content of higher alcohols, which were the precursors of esters.

Compared with that in the control (1.13 mg/L), the content of carbonyl compounds was lower in SLY-4EW (0.92 mg/L), higher in HX-13EW (1.33 mg/L), and had no significant difference in F2-24EW (1.11 mg/L) (Figure 3). The detected carbonyl compounds in wines might have negative effects on the flavor of wines, but the real effects of these compounds on wines should be further analyzed.

The odor active fermentative aroma compounds (OAV > 1) were isoamyl alcohol,1-nonanol, 3-methyl-1-pentanol, benzene ethanol, isovaleric acid, 2-methyl butyric acid, octanoic acid, ethyl butyrate, ethyl isovalerate, ethyl caproate, ethyl heptanoate, ethyl octanoate, ethyl caprate, isoamyl acetate, 2-methylbutyl acetate, phenylethyl acetate, nonanal, decanal, octanal, and phenylacetaldehyde. These results indicated that adding SLY-4E, HX-13E, and F2-24E into must could increase the content of fermentative aroma compounds, especially higher alcohols and esters, but decrease the content of fatty acids in wines.

The hierarchical clustering and heat map visualization of fermentative aroma compounds in wines implied that wines were classified into SLY-4EW/control and HX-13EW/F2-24EW, and the fermentative aroma compounds were clustered into class Ⅰ, Ⅱ, Ⅲ, and Ⅳ (Figure 4). The control was rich in class Ⅲ and class Ⅳ which contained higher alcohols, fatty acids, other esters, and phenylethyl acetate. SLY-4EW was abundant in class Ⅰ, class Ⅲ, and class IV including higher alcohols, fatty acid ethyl esters, carbonyl compounds, and acetic esters. These compounds presented fruity, floral, and bitter flavors and improved the complexity of the aroma in SLY-4EW. HX-13EW had higher contents of compounds from class Ⅰ, class II, and class Ⅲ including higher alcohols, fatty acids, acetic esters, carbonyl compounds, ethyl laurate, and diethyl succinate. F2-24EW was abundant in class Ⅰ, class Ⅱ, and class Ⅳ including fatty acid ethyl esters, acetic esters, carbonyl compounds, phenylethyl octanoate, 2-methyl butyric acid, 1-pentanol, and 4-methyl-1-pentanol. In class Ⅰ and II, the OVA of isoamyl alcohol, benzene alcohol, ethyl caprate, ethyl heptanoate, ethyl octanoate, isoamyl acetate, ethyl butyrate, ethyl caproate, and octanal were greater than 1. These compounds presented fruity, floral, and bitter flavors and improved the complexity of the aroma in SLY-4EW, F2-24EW, and HX-13EW. These results suggested that adding SLY-4E, HX-13E, and F2-24E into must produced different profiles of fermentative aroma compounds which would impart different flavor complexities on wines.

In the future, more non-*Saccharomyces* yeasts with β-glucosidases should be selected to produce different profiles of fermentative compounds. Moreover, it is also important to study the mechanism of adding different β-glucosidases from different non-*Saccharomyces* yeasts into must on the contents of fermentative aroma compounds of wines.

### 3.5. Sensory Evaluation of Wines

The sensory evaluation of wines (Figure 5) showed that the appearance had no significant difference. Compared with those in control, the scores of floral, fruity, and taste in SLY-4EW, HX-13EW, and F2-24EW were higher, while their unpleasant green flavor were lower. F2-24EW had the highest scores in floral (7.50), fruity (7.83), and taste (7.83), while HX-13EW had the lowest scores in floral (6.33), fruity (6.67), and taste (6.83). Adding β-glucosidase from yeasts could improve the fruity and floral aspects, which was also reported by Ma et al. (2017) [37], Thongekkaew et al. (2019) [12], Hu et al. (2016b) [36], and Sun et al. (2018) [10]. These results indicated that β-glucosidases from yeasts could enhance the flavor complexity and typicality of wines. The appearance of SLY-4EW, HX-13EW, and F2-24EW had no significant difference, while Wang et al. (2013) [49] reported that adding crude extracts of β-glucosidase from *T. asahii* into grape juice had a strong effect on the appearance. Part of the β-glucosidase could break the glycosidic bond, then the free anthocyanins degraded into colorless compounds, which made the wine pale [50,51,52]. The difference in the decomposition of anthocyanins by β-glucosidase might be related to the structural characteristics of substrates and properties of β-glucosidase. The effect of adding β-glucosidase from non-*Saccharomyces* yeasts into must on the appearance of wines during aging and storage should be further investigated, and the substrate specificity of β-glucosidase from different non-*Saccharomyces* yeasts should be deeply studied.

The volatile aroma compound profiles of SLY-4EW, HX-13EW, and F2-24EW were significantly different from the control. Different volatile aroma compound profiles would take different flavor characteristics on wines [53,54], while some studies found that the co-fermentation with *S. cerevisiae* and *T. delbrueckii* and *P. fermentans* could not only enhance the flavor of the wine, but also bring a vinegar or earthy flavor [37,55]. It means that there was a complex multivariate correlation between the aroma characteristics and volatile aroma compounds of wines.

## 4. Conclusions

Adding SLY-4E, F2-24E, and HX-13E into must had no negative effect on the fermentation period and the physicochemical characteristics of wines, although the maximum biomass of *S. cerevisiae* had increased. The content of terpenes, higher alcohols, and esters was increased by adding SLY-4E, F2-24E, and HX-13E, which enhanced the fruity and floral aspects of wines. Additionally, the content of C_6_ compounds was decreased which reduced the unpleasant green of wines. The content of fatty acids was decreased, which might have affected the flavor complexity of wines. Moreover, β-glucosidase from different yeast species produced different aroma compound profiles, which presented different fruity, floral, and taste aspects. However, there was no significant difference in the appearance of wines. F2-24EW had the best improvement in the floral, fruity, and taste aspects, followed by SLY-4EW and HX-13EW. These results can provide references for using β-glucosidase from different non-*Saccharomyces* yeasts to improve the flavor complexity, typicality, and quality of wines. However, the effect of the SLY-4E, F2-24E, and HX-13E on the appearance of wines during their aging and storage needs further investigation.

## Figures and Tables

**Figure 1 jof-08-01057-f001:**
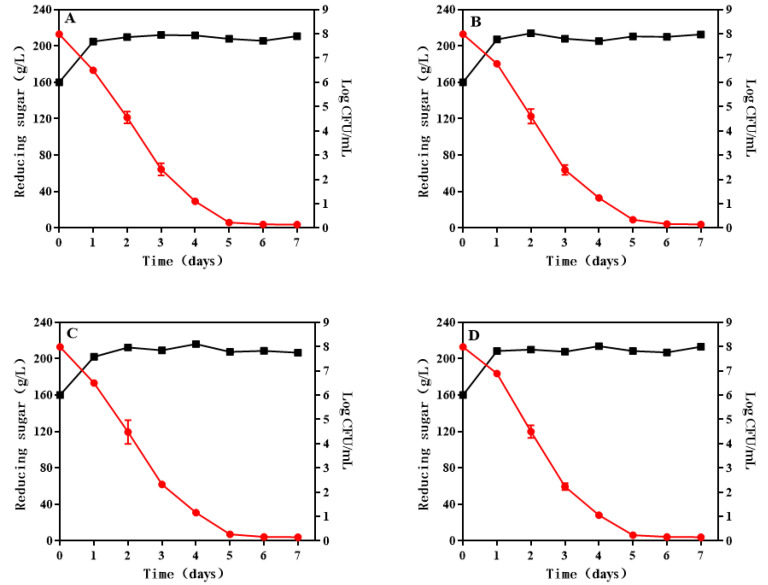
Growth and sugar consumption kinetics of *S. cerevisiae* during wine fermentation. (**A**) Control; (**B**) SLY-4EW; (**C**) HX-13EW; (**D**) F2-24EW; -■-Growth kinetics; -●-Sugar consumption kinetics.

**Figure 2 jof-08-01057-f002:**
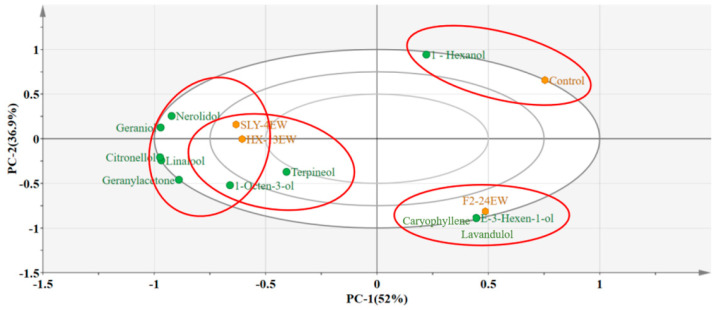
PCA of varietal aroma compounds and wines. 
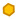
 Wines; 
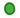
 Varietal aroma compounds.

**Figure 3 jof-08-01057-f003:**
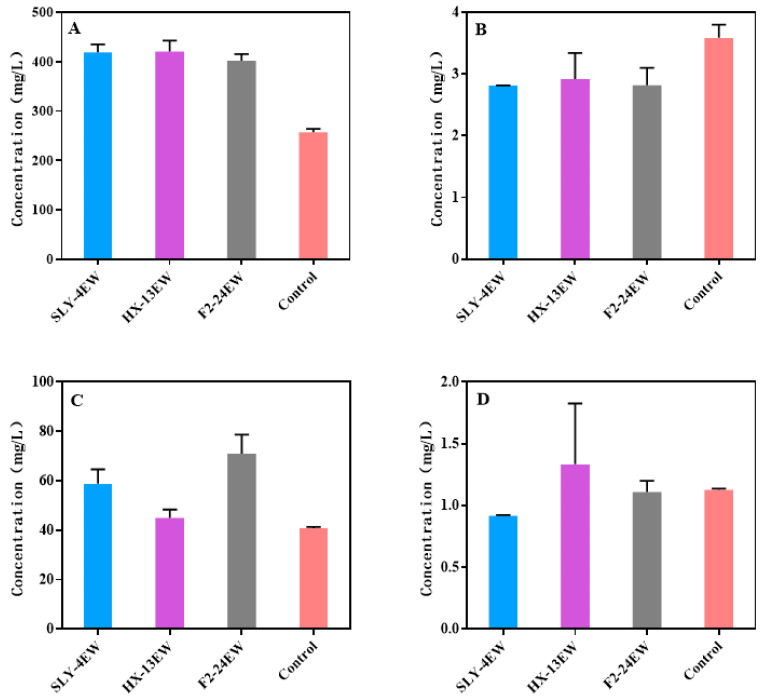
Concentration of fermentation aroma components from wines. (**A**) Higher alcohols; (**B**) Fatty acids; (**C**) Esters; (**D**) Carbonyl compounds.

**Figure 4 jof-08-01057-f004:**
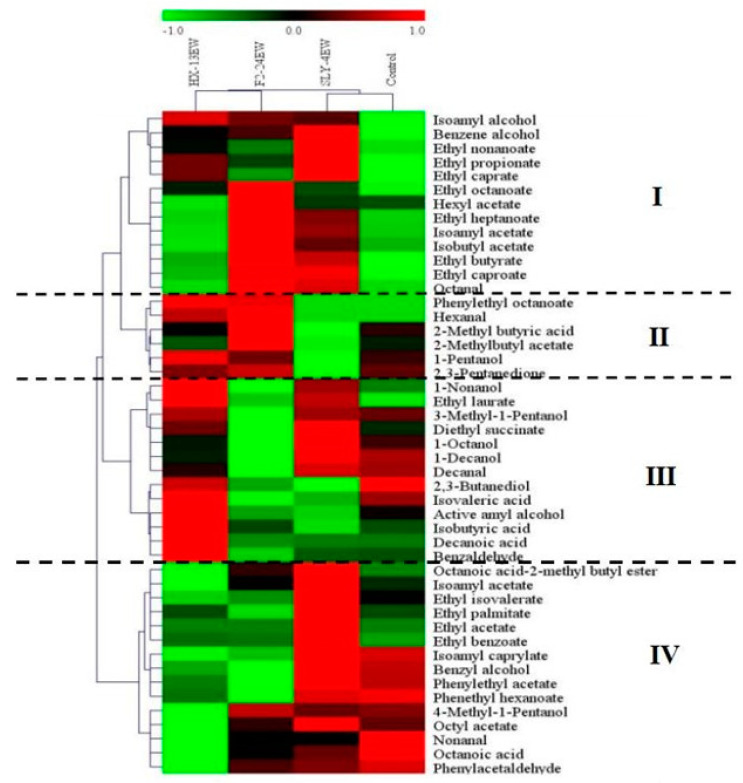
Hierarchical clustering and heat map visualization of fermentative aroma compounds from wines.

**Figure 5 jof-08-01057-f005:**
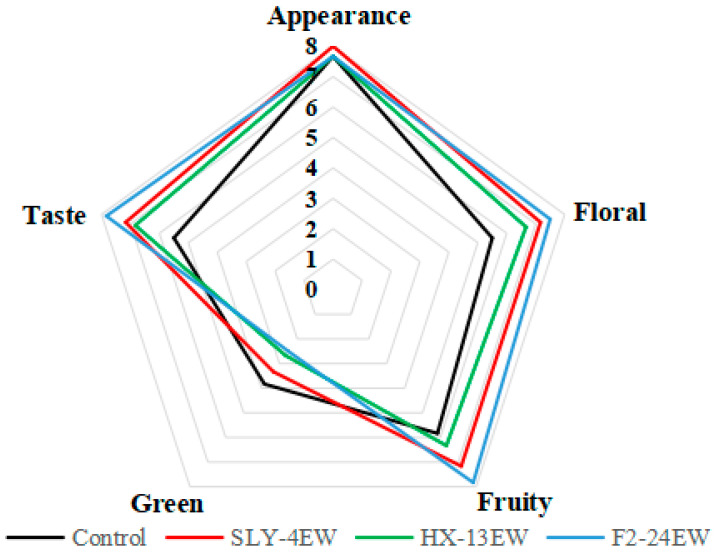
Sensory evaluation of wines.

**Table 1 jof-08-01057-t001:** Physiochemical characteristics of wines.

Wine	Fermentation Period (d)	Residual Sugar (g/L)	Alcohol (%, *v*/*v*)	Total Acid (g/L)	Volatile Acid (g/L)
Control	7	3.83 ± 0.02 ^a^	11.62 ± 0.15 ^a^	6.66 ± 0.13 ^a^	0.29 ± 0.02 ^a^
SLY-4EW	7	4.00 ± 0.08 ^a^	11.99 ± 1.02 ^a^	6.75 ± 0.27 ^a^	0.28 ± 0.01 ^a^
HX-13EW	7	3.93 ± 0.05 ^a^	11.95 ± 0.15 ^a^	6.56 ± 0.27 ^a^	0.27 ± 0.01 ^a^
F2-24EW	7	3.98 ± 0.16 ^a^	11.63 ± 0.59 ^a^	6.66 ± 0.40 ^a^	0.27 ± 0.01 ^a^

Notes: Subscripts in the same column indicate significant differences (α = 0.05).

**Table 2 jof-08-01057-t002:** Concentration of volatile aroma compounds in wines (mg/L).

Compounds	Wines				Odor Threshold	OAV	Sensory Description
	SLY-4EW	HX-13EW	F2-24EW	Control			
1-Hexanol	0.91 ± 0.08 ^ab^	0.84 ± 0.04 ^b^	0.79 ± 0.04 ^b^	0.98 ± 0.07 ^a^	8 [21]	0.1–1	Herbaceous, grass [21]
E-3-Hexen-1-ol	-	-	0.03 ± 0.00 ^a^	-	0.4 [22]	<0.1	Herbaceous, grass [21]
**C_6_ compounds**	0.91 ± 0.08 ^ab^	0.84 ± 0.04 ^b^	0.82 ± 0.04 ^b^	0.98 ± 0.07 ^a^			
Linalool	0.40 ± 0.03 ^a^	0.39 ± 0.01 ^a^	0.30 ± 0.02 ^b^	0.21 ± 0.05 ^c^	0.1 [22]	>1	Muscat, flowery, fruity [23]
Citronellol	0.24 ± 0.00 ^a^	0.23 ± 0.03 ^a^	0.14 ± 0.03 ^b^	0.08 ± 0.01 ^c^	0.1 [23]	>1	Green lemon [21]
1-Octen-3-ol	0.20 ± 0.03 ^a^	0.14 ± 0.04 ^b^	0.16 ± 0.04 ^ab^	0.08 ± 0.01 ^c^	0.02 [23]	>1	Mushroom [23]
Geranylacetone	0.05 ± 0.02 ^a^	0.06 ± 0.03 ^a^	0.04 ± 0.00 ^a^	-	0.06 [24]	0.1–1	Flowery [24]
Nerolidol	0.12 ± 0.01 ^a^	0.10 ± 0.01 ^b^	0.06 ± 0.00 ^c^	0.07 ± 0.00 ^c^	0.7 [23]	0.1–1	Roses, apple, orange [23]
Terpineol	-	0.06 ± 0.01 ^a^	0.02 ± 0.02 ^b^	-	0.25 [25]	0.1–1	Flowery, piny [26]
Geraniol	0.06 ± 0.01 ^b^	0.07 ± 0.00 ^a^	-	-	0.03 [25]	>1	Roses [25]
Caryophyllene	-	-	0.08 ± 0.01 ^a^	-	0.064	>1	Spicy, woody, orange
Lavandulol	-	-	0.02 ± 0.000 ^a^	-			-
**Terpenes**	1.07 ± 0.11 ^a^	1.04 ± 0.13 ^a^	0.82 ± 0.08 ^b^	0.43 ± 0.07 ^c^			
**Varietal aroma compounds**	1.98 ± 0.19 ^a^	1.88 ± 0.17 ^a^	1.64 ± 0.12 ^a^	1.41 ± 0.13 ^b^			
Isoamyl alcohol	211.78 ± 15.74 ^b^	238.09 ± 1.14 ^a^	217.26 ± 6.67 ^b^	114.37 ± 1.24 ^c^	30 [27]	>1	Whiskey, malt, burnt [27]
2,3-Butanediol	0.93 ± 0.31 ^a^	1.15 ± 0.35 ^a^	0.98 ± 0.04 ^a^	1.19 ± 0.12 ^a^	120 [28]	<0.1	Butter, creamy [28]
1-Pentanol	0.02 ± 0.00 ^d^	0.10 ± 0.00 ^a^	0.09 ± 0.01 ^b^	0.08 ± 0.01 ^c^	80	<0.1	Mellow, astringency
1-Octanol	0.85 ± 0.06 ^a^	0.70 ± 0.21 ^ab^	0.56 ± 0.06 ^b^	0.74 ± 0.15 ^ab^	0.9 [29]	0.1–1	Flesh orange, rose, sweetherb [29]
1-Nonanol	0.71 ± 0.36 ^ab^	0.82 ± 0.04 ^a^	0.39 ± 0.06 ^b^	0.51 ± 0.02 ^b^	0.015 [30]	>1	Orange [30]
1-Decanol	0.19 ± 0.07 ^a^	0.16 ± 0.07 ^a^	0.13 ± 0.00 ^a^	0.18 ± 0.01 ^a^	0.4 [27]	0.1–1	Flowery [27]
4-Methyl-1-Pentanol	0.47 ± 0.01 ^a^	0.41 ± 0.03 ^b^	0.48 ± 0.01 ^a^	0.47 ± 0.08 ^a^	50 [31]	<0.1	Almonds, toast [31]
3-Methyl-1-Pentanol	0.98 ± 0.17 ^a^	0.98 ± 0.07 ^a^	0.85 ± 0.02 ^a^	0.96 ± 0.14 ^a^	0.5 [31]	>1	Soil, mushroom [31]
Active amyl alcohol	48.44 ± 3.11^b^	52.32 ± 0.77 ^a^	48.76 ± 1.32 ^b^	50.01 ± 0.44 ^ab^	65 [32]	0.1–1	Hetero alcohol, almond [32]
Benzyl alcohol	0.24 ± 0.01 ^a^	0.17 ± 0.00 ^b^	0.16 ± 0.03 ^b^	0.23 ± 0.05 ^a^	200 [29]	<0.1	Almond [29]
Benzene alcohol	154.28 ± 2.76 ^a^	125.79 ± 23.35 ^b^	132.20 ± 5.58 ^b^	88.23 ± 9.09 ^c^	7.5 [29]	>1	Soil, mushroom [29]
**Higher alcohols**	418.89 ± 22.60 ^a^	420.71 ± 26.03 ^a^	401.83 ± 13.79 ^a^	256.97 ± 11.36 ^b^			
Isobutyric acid	0.04 ± 0.00 ^c^	0.13 ± 0.04 ^b^	0.28 ± 0.01 ^a^	0.27 ± 0.04 ^a^	2.3 [25]	0.1–1	Cheese, rancid [25]
Isovaleric acid	0.77 ± 0.04 ^a^	0.97 ± 0.30 ^a^	0.74 ± 0.08 ^a^	0.90 ± 0.04 ^a^	0.03 [25]	>1	Fatty [25]
2-Methyl butyric acid	0.66 ± 0.01 ^b^	0.97 ± 0.08 ^a^	0.71 ± 0.28 ^b^	0.69 ± 0.06 ^b^	0.033 [21]	>1	Cheese [21]
Octanoic acid	1.10 ± 0.04 ^b^	0.69 ± 0.04 ^d^	0.93 ± 0.05 ^c^	1.46 ± 0.23 ^a^	0.5 [22]	>1	Cheese, rancid [22]
Decanoic acid	0.25 ± 0.00 ^a^	0.16 ± 0.03 ^b^	0.16 ± 0.02 ^b^	0.27 ± 0.01 ^a^	1 [25]	0.1–1	Fatty, unpleasant [25]
**Fatty acids**	2.81 ± 0.09^b^	2.92 ± 0.50 ^b^	2.82 ± 0.44 ^b^	3.58 ± 0.38 ^a^			
Ethyl acetate	0.10 ± 0.02 ^a^	-	-	-	7.5 [29]	<0.1	Fruity, sweet taste [29]
Ethyl propionate	0.36 ± 0.11 ^a^	0.30 ± 0.03 ^a^	0.28 ± 0.03 ^a^	0.25 ± 0.08 ^a^	1.8 [23]	0.1–1	Pineapples, bananas, apples [23]
Ethyl butyrate	0.77 ± 0.43 ^a^	0.48 ± 0.05 ^a^	0.83 ± 0.54 ^a^	0.41 ± 0.14 ^a^	0.02 [27]	>1	Strawberries, apples, bananas [27]
Ethyl isovalerate	1.05 ± 0.42 ^a^	-	0.11 ± 0.01 ^bc^	0.38 ± 0.02 ^b^	0.003	>1	Bananas, fruity
Ethyl caproate	21.46 ± 2.68 ^a^	9.26 ± 0.80 ^b^	21.58 ± 4.69 ^a^	7.27 ± 1.28 ^b^	0.014 [27]	>1	Green apples, fennel [27]
Ethyl heptanoate	0.11 ± 0.01 ^a^	0.07 ± 0.01 ^b^	0.13 ± 0.05 ^a^	0.07 ± 0.01 ^b^	0.002 [27]	>1	Sweet, strawberries, bananas [27]
Diethyl succinate	0.31 ± 0.10 ^a^	0.29 ± 0.01 ^ab^	0.21 ± 0.01 ^b^	0.26 ± 0.09 ^ab^	200 [27]	<0.1	Fruity, melons [27]
Ethyl octanoate	9.80 ± 0.27 ^b^	10.48 ± 1.19 ^b^	16.60 ± 0.61 ^a^	7.10 ± 0.19 ^c^	0.25 [27]	>1	Fruity [27]
Ethyl nonanoate	0.44 ± 0.24 ^a^	0.29 ± 0.03 ^ab^	0.25 ± 0.03 ^b^	0.21 ± 0.06 ^b^	1.3 [27]	0.1–1	Flowery, fruity [27]
Ethyl caprate	7.29 ± 0.28 ^a^	5.79 ± 0.60 ^b^	4.01 ± 0.61 ^c^	3.27 ± 0.57 ^d^	0.2 [33]	>1	Apples, flowery [29]
Ethyl laurate	0.53 ± 0.15 ^a^	0.57 ± 0.05 ^a^	0.35 ± 0.04 ^b^	0.34 ± 0.07 ^b^	1.5 [34]	0.1–1	Fruity, fatty [34]
Ethyl palmitate	0.15 ± 0.01 ^a^	0.08 ± 0.05 ^b^	0.06 ± 0.01 ^c^	0.08 ± 0.00 ^b^	1.5 [34]	<0.1	Fatty, fruity, sweet [34]
**Fatty acid ethyl esters**	42.32 ± 4.72 ^a^	27.60 ± 2.82 ^b^	44.40 ± 6.62 ^a^	19.63 ± 2.52 ^c^			
Isoamyl acetate	6.82 ± 0.44 ^b^	3.44 ± 0.44 ^d^	8.43 ± 0.32 ^a^	3.64 ± 0.08 ^c^	0.2 [27]	>1	Green apples, bananas [27]
2-Methylbutyl acetate	-	1.61 ± 0.48 ^b^	6.33 ± 0.38 ^a^	2.16 ± 0.35 ^b^	0.16 [27]	>1	Bananas, fruity [27]
Isobutyl acetate	0.29 ± 0.14 ^a^	0.19 ± 0.07 ^a^	0.36 ± 0.23 ^a^	0.21 ± 0.02 ^a^	1.6 [27]	0.1–1	Bananas [29]
Hexyl acetate	0.44 ± 0.00 ^b^	0.32 ± 0.09 ^b^	0.73 ± 0.47 ^a^	0.42 ± 0.07 ^ab^	1.5 [27]	0.1–1	Fruity, pear, cherry [27]
Octyl acetate	0.20 ± 0.09 ^a^	-	0.13 ± 0.04 ^a^	0.15 ± 0.01 ^a^			-
Phenylethyl acetate	13.89 ± 0.19 ^a^	10.761 ± 0.241 ^c^	9.39 ± 0.21 ^d^	13.17 ± 0.42 ^b^	0.65 [34]	>1	Fruity, flowery [34]
**Acetic esters**	21.63 ± 0.86 ^ab^	16.33 ± 1.31 ^b^	25.36 ± 1.65 ^a^	19.74 ± 0.94 ^b^			
Octanoic acid-2-methyl butyl ester	0.30 ± 0.00 ^a^	0.12 ± 0.00 ^d^	0.21 ± 0.01 ^b^	0.16 ± 0.02 ^c^			
Isoamyl acetate	0.35 ± 0.08 ^a^	0.13 ± 0.03 ^c^	0.23 ± 0.11 ^b^	0.21 ± 0.02 ^bc^	1 [25]	0.1–1	Apples, pineapples [25]
Isoamyl caprylate	0.55 ± 0.14 ^a^	0.32 ± 0.07 ^b^	0.34 ± 0.06 ^b^	0.52 ± 0.04 ^a^	0.125 [25]	0.1–1	Fruity, cheese [25]
Ethyl benzoate	0.34 ± 0.02 ^a^	0.21 ± 0.01 ^b^	0.21 ± 0.00 ^b^	0.20 ± 0.00 ^b^			
Phenethyl hexanoate	0.34 ± 0.03 ^a^	0.23 ± 0.02 ^b^	0.17 ± 0.01 ^c^	0.35 ± 0.06 ^a^			
Phenethyl octanoate	-	0.03 ± 0.01 ^a^	0.03 ± 0.01 ^a^	-			
**Other esters**	1.88 ± 0.27 ^a^	1.03 ± 0.14 ^c^	1.18 ± 0.27 ^c^	1.43 ± 0.15 ^b^			
**Esters**	58.82 ± 5.81 ^b^	44.96 ± 1.68 ^c^	70.95 ± 7.77 ^a^	40.80 ± 4.17 ^c^			
Nonanal	0.11 ± 0.01 ^b^	0.08 ± 0.02 ^c^	0.11 ± 0.02 ^b^	0.14 ± 0.02 ^a^	0.015 [27]	>1	Green, spicy [27]
Decanal	0.07 ± 0.01 ^a^	0.06 ± 0.00 ^b^	0.03 ± 0.00 ^c^	0.06 ± 0.00 ^a^	0.001	>1	Flowery
Octanal	0.06 ± 0.01 ^b^	-	0.06 ± 0.01 ^a^	-	0.001	>1	Bitter, lemons
Hexanal	-	0.01 ± 0.01 ^a^	0.01 ± 0.01 ^a^	-			
2,3-Pentanedione	0.16 ± 0.08 ^b^	0.41 ± 0.02 ^a^	0.44 ± 0.07 ^a^	0.40 ± 0.05 ^a^	2 [21]	0.1–1	Pecans [21]
Benzaldehyde	0.40 ± 0.10 ^a^	0.68 ± 0.48 ^a^	0.327 ± 0.024 ^b^	0.40 ± 0.02 ^a^	2 [21]	0.1–1	Toasted almonds [21]
Phenylacetaldehyde	0.12 ± 0.00 ^a^	0.10 ± 0.01 ^a^	0.12 ± 0.04 ^a^	0.12 ± 0.02 ^a^	0.005 [25]	>1	Flowers, roses, honey [25]
**Carbonyl compounds**	0.92 ± 0.19 ^b^	1.33 ± 0.53 ^a^	1.11 ± 0.16 ^ab^	1.13 ± 0.10 ^ab^			
**Fermentation aroma**	488.45 ± 28.73 ^a^	469.92 ± 31.32 ^a^	476.71 ± 22.85 ^a^	302.48 ± 15.44 ^b^			

Note: Subscripts in the same line indicate significant difference (α = 0.05); “-” means the compound is not detected.

## Data Availability

Not applicable.

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
