# Peer review of "Effects of Purified β-Glucosidases from Issatchenkia terricola, Pichia kudriavzevii, Metschnikowia pulcherrima on the Flavor Complexity and Typicality of Wines"

_jof, 2022, doi:10.3390/jof8101057_

Round 1
Reviewer 1 Report
The authors emphasize that beta-glucosidases enhance the typicality of wines, it would be better explained in which sense. In fact, the same idea is repeated several times in the first paragraphs of the manuscript.
What is typicality for you, and specially in Cabernet sauvignon to conclude if your strains contributed to it.
The sensory descriptors analysed should be better explained. For example, green is a good or bad property? Later you described it is unpleasant but in Fig5 is not clear. Is green descriptor in mouth or smell? A high number is good or bad? Is this related with the typicality that you mentioned? What is in aspect? Please, provide a brief explanation f these characteristics.
Line 61: specify geographical origin of the strains
Line 66: pH of medium was controlled?
Line 77: residual sugar could be total sugar or reducing sugars?
Line 86: do you mean reducing sugars?
Line 87 and 90: the methods should be listed in the references
Line 112: the sensory analysis was made using visual, smell and mouth aspects or just smell?
Line 130: Is this difference significant? Because I cannot distinguish any evident change in the graphic.
Line 217: extracts of
Line 287: to produce
Reviewer 2 Report
Dear authors:
Your paper present a work that, although not novel, increases the knowledge about the addition of non-conventional non-Saccharomyces B-glucosidases fermentation, which can be relevant to improve the quality and differentiation of wines. This is important in Denominations of Origin and wineries that intend to use indigenous yeasts to improve their wines with unique and typical identity brands; while maintaining biogeographical patterns of yeasts with potential use of their B-glucosidases that differentiate them from the rest of the market. The paper is well thought out, executed and written, however, in my opinion, some modifications, clarifications and minor changes are required to improve the quality of the manuscript to publish in J. Fungi 2022:
Throughout the document there are misplaced commas and spaces, or a symbol that is not a comma, etc. correct this.
136-139 briefly discuss what a higher biomass affects: Improve fermentation time? , ensure fermentation?.
157 Mention is made of table 2. It should appear near this mention.
188 and 207 Indicate if all compounds were used or if only those with OAV>1 were used and why.
221 Proper content of higher alcohols (<300 221 mg/L) would bring fruity and floral flavor to wines [31-33]. However, they exceeded 400 mg/L which can be counterproductive: discuss this.
263-264 I suggest that since it is already indicated in the figure 4, define here compounds in each class in parentheses and in the following paragraphs (265-280) just only indicate the class for each wine.
265-280 In relation to the previous suggestion it would be convenient to indicate the class for each wine thus avoiding repeating the compounds in each class so that it is clearer to read. Also discuss which wine-groups may be more favorable according to quality or differentiation: for example, more favorable could be the group with a higher quantity of esters or the wine with a higher concentration of some compounds.
296 and 300 These two phrases should be unified as they repeat the same concept.
311-314 Perhaps it would be convenient (if they are available), to add more features in the sensory analysis that is a bit lacking in parameters.
315-316 A Pearson correlation between aromatic compounds and sensory parameters could be useful.
326 The article "the the" is repeated at the end of the sentence.
Congratulations for the work done and good luck with your new research in favor of expanding knowledge.
